# The Influence of Forward Osmosis Module Configuration on Nutrients Removal and Microalgae Harvesting in Osmotic Photobioreactor

**DOI:** 10.3390/membranes12090892

**Published:** 2022-09-16

**Authors:** Mathieu Larronde-Larretche, Xue Jin

**Affiliations:** 1Keon Research LLC, 2151 N Northlake Way b, Seattle, WA 98103, USA; 2School of Chemical Engineering, Biological Engineering & Environmental Engineering, Oregon State University, Corvallis, OR 97331, USA

**Keywords:** wastewater treatment, microalgae dewatering, membrane photobioreactor, fouling, *Chlorella vulgaris*

## Abstract

Microalgae have attracted great interest recently due to their potential for nutrients removal from wastewater, renewable biodiesel production and bioactive compounds extraction. However, one major challenge in microalgal bioremediation and the algal biofuel process is the high energy cost of separating microalgae from water. Our previous studies demonstrated that forward osmosis (FO) is a promising technology for microalgae harvesting and dewatering due to its low energy consumption and easy fouling control. In the present study, two FO module configurations (side-stream and submerged) were integrated with microalgae (*C. vulgaris*) photobioreactor (PBR) in order to evaluate the system performance, including nutrients removal, algae harvesting efficiency and membrane fouling. After 7 days of operation, both systems showed effective nutrients removal. A total of 92.9%, 100% and 98.7% of PO_4_-P, NH_3_-N and TN were removed in the PBR integrated with the submerged FO module, and 82%, 96% and 94.8% of PO_4_-P, NH_3_-N and TN were removed in the PBR integrated with the side-stream FO module. The better nutrients removal efficiency is attributed to the greater algae biomass in the submerged FO-PBR where in situ biomass dewatering was conducted. The side-stream FO module showed more severe permeate flux loss and biomass loss (less dewatering efficiency) due to algae deposition onto the membrane. This is likely caused by the higher initial water flux associated with the side-stream FO configuration, resulting in more foulants being transported to the membrane surface. However, the side-stream FO module showed better fouling mitigation by simple hydraulic flushing than the submerged FO module, which is not convenient for conducting cleaning without interrupting the PBR operation. Taken together, our results suggest that side-stream FO configuration may provide a viable way to integrate with PBR for a microalgae-based treatment. The present work provides novel insights into the efficient operation of a FO-PBR for more sustainable wastewater treatment and effective microalgae harvesting.

## 1. Introduction

It is estimated that the world produces around 359 billion cubic meters of wastewater annually, of which more than 48% is untreated [1]. The discharge of untreated wastewater into the natural water environment can lead to serious environmental problems, particularly deteriorating water quality and damaging aquatic ecosystems [2]. Moreover, insufficient wastewater treatment implies a loss of nutrients [3]. The energy required to treat wastewater is high (0.8–2.1 kWh/m^3^) [4,5]. In the United States, the water industry is a very large user of energy. Water treatment accounts for ~2% of energy consumption, adding over 45 million tons of greenhouse gases annually [6]. The increasing energy demand has, in large part, been driven by increasingly stringent environmental regulations. Thus, there has been a push towards advanced treatment technologies, such as membrane bioreactors, that can guarantee very high-quality effluent but at the price of increased energy usage. In the long term, wastewater treatment will be far more environmentally and economically sustainable if we can, not only reduce the energy associated with pollution control, but also recover valuable resources from the wastewater. Novel technologies that can achieve this will have a distinct advantage in a ready and expanding global market for water supply and remediation.

In recent years, the application of microalgae in wastewater treatment has attracted increasing attention as a promising strategy for nutrients removal, greenhouse gas abatement and biofuel production. The algae-based approach is environmentally sustainable as it depends on the principle of the natural ecosystem [7]. Microalgae uptake nutrients from wastewater and accumulate significant amounts of value-added lipids in their cells, showing great potential for biodiesel production [8,9]. Several microalgae species including *Chlorella* and *Scenedesmus* have been considered as promising candidates for wastewater treatment due to their great nutrients removal efficiency and excellent tolerance to different wastewater conditions [8,10]. Despite the promise, one technical challenge remaining to be overcome is the high-energy cost of algae harvesting and dewatering that accounts for 20–30% of the total operating cost [11]. In order to make microalgae-based technology economically feasible, algae harvesting must be achieved based on a low-energy method [8].

Forward osmosis (FO) is a natural process by which clean water passes from dirty feed water towards a salt ‘draw’ solution with higher osmotic pressure when the two solutions are separated by a semipermeable membrane [12]. It demonstrates unparalleled advantages of low energy consumption, superb solute retention, and potentially low fouling tendency [13]. In our previous studies, a crossflow FO system was investigated as an economical and sustainable alternative for algae harvesting. We found that the species of algae and draw solution chemistry play an important role in the overall filtration performance [14,15]. Compared to *Scenedesmus obliquus* and *Chlamydomonas reinhardtii*, *Chlorella vulgaris* was the most suitable species to be harvested by FO with outstanding biomass recovery and negligible membrane fouling [15]. In 2016, Praveen and Loh [16] proposed to integrate FO and a photobioreactor (PBR) for continuous nutrient removal from wastewater using *C. vulgaris*. In this study, 95% NH_3_-N removal, 53% NO_2_-N removal and 89% PO_4_-P removal were achieved. However, microalgae exhibited a high tendency to foul membrane in the submerged FO-PBR.

The membrane module design and hydrodynamic control are critical aspects for osmotic membrane processes [17]. Good module design and hydrodynamic control can significantly enhance the separation performance, reduce the risk of fouling, and ensure successful implementation of the proposed technology. The module and system must be designed in such a way as to allow (1) maximized system performance, (2) simplicity in implementation and system integration, and (3) ease of scale up. FO-PBR systems can be classified into submerged and side-stream configurations. To be more cost-effective and simple, submerged modules may be preferred as they require low energy usage due to the elimination of feedwater recirculation [17]. However, the difficulties in maintenance and salts accumulation inside the reactor, which affect both microalgae growth and nutrients removal efficiency through osmotic stress [18], are common challenges with the submerged configuration [19]. In the side-stream configuration, the FO module is external to the PBR. Both feed and draw solutions are circulated tangentially across the membrane surface. Some attractive features of the side-stream module include better fouling control by high crossflow velocity and easier maintenance [20]. In order to better understand the efficiency of FO-PBR, especially for long-term operation, a detailed and systematic comparison between the two configurations is needed. However, to the best of our knowledge, the impact of FO-PBR configurations on nutrients removal, algae biomass dewatering and membrane fouling control has yet to be reported. Hence, this study focuses on this gap and aims to experimentally compare the nutrients removal efficiency, biomass dewatering, water flux and fouling behavior between submerged and side-stream configurations. We believe our results will provide critical insights into the efficient operation of FO-PBR for more sustainable wastewater treatment and effective microalgae harvesting.

## 2. Materials and Methods

### 2.1. Microalgae Strain and Pre-Cultivation

In addition to its excellent potential for wastewater treatment and CO_2_ capture as well as high lipid productivity [9,21,22], *C. vulgaris* has been reported to be the most suitable microalgae species to be dewatered by FO [15]. Thus, *C. vulgaris* was selected in our study. The pure culture of *C. vulgaris* was purchased from the Culture Collection of Algae and Protozoa (CCAP, Oban, UK) and pre-cultivated in a modified BG-11 medium following the previous protocol [15,23]. Cultures were grown at room temperature (25 ± 1 °C) and illuminated by fluorescent light tubes at 100 μmol photons/m^2^∙s. The pH was monitored and maintained at 7 ± 0.5. The algae growth phase was monitored daily by optical density measurement with a spectrophotometer (Helios Zeta, Thermo Scientific, Inchinnan, Scotland) at a 435 nm wavelength [24]. The microalgae suspension was harvested at the end of the exponential phase before the batch cultivation in 7.2 L PBR was integrated with FO modules (FO-PBR, details are described in Section 2.4).

### 2.2. Synthetic Wastewater and Draw Solution Chemistry

The synthetic wastewater used for algal cultivation in FO-PBR was prepared by dissolving ACS reagent grade chemicals (Sigma-Aldrich, Gillingham, UK) in deionized water. Its composition is provided in Table 1 [25]. The synthetic wastewater was autoclaved prior to each experiment in order to avoid contamination. A commercial sea salt was used to prepare the draw solution (DS) in FO experiments. The concentration of sea salt was 70 g/L to mimic the salinity of brine from a typical reverse osmosis (RO) desalination plant [26]. Its composition was reported in our previous study [14,15].

### 2.3. FO Membrane

All FO experiments were conducted with flat-sheet thin-film composite (TFC) membrane coupons (Porifera, Hayward, CA, USA). The membrane has an active layer made of polyamide, coated on top of a polysulfide support layer with an embedded woven mesh [27]. The membrane active layer exhibited a negative zeta potential at neutral pH due to the abundant carboxyl groups of the polyamide active layer [28]. Detailed membrane properties have been reported in a previous study [27,29].

### 2.4. Experimental Setup and Protocols

#### 2.4.1. Experimental Setup

Laboratory-scale PBRs were coupled with submerged and side-stream FO modules to investigate the effects of the FO testing configuration on nutrients removal, microalgae harvesting, water flux, and membrane fouling. The flat panel PBR was designed for the continuous growth of microalgae in synthetic wastewater. The PBR is made of transparent polymethyl methacrylate (PMMA) and has an effective volume of 7.2 L (20 cm length × 12 cm width × 30 cm height). Its bottom comprises 14 pin holes designed for the injection and bubbling of air to provide inorganic carbon to the microalgae. The PBR was illuminated from the sides using a fluorescent light. Prior to each experiment, the PBR was sterilized by filling it with peroxyacetic acid (1%) and rinsed three times with deionized water in order to avoid contamination from other microorganisms.

Two plate-and-frame FO membrane modules (submerged and side-stream) were integrated with the PBR to harvest the microalgae biomass. The two FO modules were designed to have the same effective membrane area (200 cm^2^). Figure 1 shows the schematic of the FO-PBR setups with different FO module configurations. In both configurations, a DS tank was placed on a digital scale (Denver Instrument, Bohemia, NY, USA) which was interfaced with an automatic data acquisition system to record weight changes and determine permeate water flux. For the side-stream module, a flat sheet membrane coupon was housed in a crossflow membrane cell, similar to that described in our previous studies [14,15]. A counter-current flow was used to circulate both feed and draw solutions on both sides of the membrane at a crossflow velocity of 9.6 cm/s. The membrane active layer faced the feed solution, which was well mixed by a magnetic stirrer to prevent microalgae sedimentation. A mesh spacer was placed on the DS side of the membrane to provide mechanical strength and promote turbulence. In this configuration, membrane fouling can be controlled by operating with high crossflow velocity that generates shear stress on the membrane surface [30]. The submerged module was immersed in the PBR tank for osmotic filtration. Two membrane coupons were placed in the module and separated by a piece of mesh spacer. Each membrane coupon had an effective area of 100 cm^2^. The support layer of both membrane coupons faced toward the inside, and the DS was circulated through the spacer. The FO module was designed in such a way that the active layer faced the algae suspension and the support layer faced the DS. In the submerged configuration, membrane fouling can be controlled by an air scour that generates shear stress on the surface of the membrane active layer.

#### 2.4.2. Osmotic PBR Operation with Synthetic Wastewater

Experiments for nutrient removal and algae harvesting with osmotic PBR were carried out in four stages: (1) batch algae cultivation, (2) semi-continuous algae cultivation, (3) algae dewatering and (4) membrane cleaning. First, *C. vulgaris* cells were cultured in 5 L of synthetic wastewater for 7 days in the PBR (batch mode) until the algae concentration reached 0.6 g/L (dry cell weight biomass). Second, the PBR was switched to semi-continuous mode from Day 8. The feed pump introduced synthetic wastewater into the PBR continuously with a flowrate of 1.1 L/day. Before each algae dewatering experiment, 100 mL of the water sample was taken from the PBR for further analysis and 6 L of algae suspension left in the reactor. The PBR performance was monitored by taking samples for conductivity, nutrients and algae biomass concentration analysis on a daily basis.

After each 24 h of PBR operation with continuous influent, algae dewatering was conducted with two FO module configurations (submerged and side-stream) in a semi-batch mode. Differences in water flux, membrane fouling and algae harvesting efficiency between the two configurations were evaluated. With the side-stream FO module, algae dewatering was initiated with 1 L of algae suspension that was withdrawn from the PBR. The FO process was interrupted when 750 mL of permeate was extracted from the feed to the DS (the concentration factor reached 4). Samples were collected from both feed and DS tanks for further analysis. At the end of the algae dewatering experiments, both feed and DS tanks were emptied and the membrane system was rinsed with deionized water at a crossflow velocity of 19.2 cm/s for 30 min. After rinsing, the membrane module was filled with deionized water and left overnight. On the next day, the membrane module was emptied. Then the FO process was continued with the next batch with a fresh DS and the algae suspension freshly collected from the PBR.

The submerged FO module was immersed in the PBR tank (with 6 L of algae suspension) prior to the first dewatering experiment. A peristaltic pump (Masterflex L/S, Cole Palmer, Vernon Hills, IL, USA) was used to recirculate the DS through the channel within the submerged module until 750 mL of water permeated the membrane (concentration factor of 1.14), leaving 5.25 L of algae suspension inside the PBR. Some 250 mL of the concentrated biomass was then withdrawn from the reactor in order to match the volume of the biomass removed from the PBR with the side-stream FO module. Next, the DS tank was emptied, and the DS channel was rinsed with deionized water. The FO module was then left inside the PBR overnight. On the next day, the FO process was continued with the next batch. To quantify the permeate flux loss caused by algae fouling, baseline FO experiments were also conducted under identical conditions to the corresponding algae FO dewatering experiments, except that no algae biomass was added into the feed solution [14]. Permeate flux loss caused by algae fouling was determined by:(1)$$\Delta J_{w}=1-\frac{J_{w,a}}{J_{w,b}}$$
where Δ*J_w_* is the normalized water flux loss, $J_{w,a}$ and $J_{w,b}$ are the water flux in the algae dewatering test and baseline test at a specific permeate volume, respectively. To determine the algae dewatering efficiency of the FO, the algae biomass concentration in the feed tank was measured. The algae dewatering efficiency was calculated by
(2)$$R=\frac{X_{f}}{X_{i}\cdot CF}$$
where *R* is the FO dewatering efficiency (or recovery rate), and *X_i_* and *X_f_* are the algae biomass concentration in the feed tank before and after FO dewatering experiments, respectively. *CF* is the concentration factor.

### 2.5. Sampling, Analytical Methods and Calculation

Samples were collected from the FO-PBR every day to determine the efficiencies of algae biomass harvesting and nutrients removal. The algae biomass concentration in the FO feed tank was determined by measuring the dry weight. In brief, 10 mL of algae suspension was filtered using a membrane with a pore size of 0.45 μm, followed by being dried at 105 °C for 2 h and then weighed. The dry biomass was calculated based on the filter paper weight change between before and after filtration [31]. The concentrations of NH_4_^+^, NO_2_^−^, NO_3_^−^ and PO_4_^3−^ were measured by ion chromatography with conductivity detection (Metrohm AG, Ionenstrasse, Switzerland) in the PBR and the feed tank of the FO (in the case of the side-stream configuration). Due to the high salinity of DS samples, nutrients concentrations in the DS were measured using an AQUAfast™ colorimeter AQ3700 (Thermo Fisher Scientific, Inchinnan, UK). Analyses were performed in triplicate and the results were given as the mean values. Before the nutrients concentration analysis, all samples were filtered with 0.2 μm cellulose acetate filter. Water conductivity in the PBR was measured with a conductivity meter (Ultrameter II, Myron L Company, Carlsbad, CA, USA).

## 3. Results and Discussion

### 3.1. Salt Accumulation

Due to the draw solutes back diffusion through the FO membrane and great solute rejection by the membrane, salts gradually accumulate inside the submerged osmotic bioreactor [32,33]. The high bioreactor salinity may inhibit algae cells growth and reduce the microbial activity of *C. vulgaris* which is freshwater microalgae and does not have high salt tolerance [34]. Thus, the gradual increase in the salinity of the submerged osmotic PBR is a concern. Figure 2 shows salinity profiles of the PBR integrated with different FO configurations over time. On Day 1, the PBR fed with continuous wastewater started to integrate with the FO module. With the side-stream FO module, salt concentration stabilized at around 178 mg/L. In contrast, the salinity in the PBR with the submerged FO module increased gradually and monotonically, reaching over 636 mg/L on Day 7. In addition to the potential harmful impacts on algae growth and metabolism, the enhanced salinity inside the PBR will also affect FO performance due to the reduced osmotic driving force [35]. We will discuss nutrients removal efficiency, biomass harvesting, the water flux and fouling behavior with different FO configurations in the following sections.

### 3.2. Biomass Analysis

Figure 3 shows the changes in the microalgae biomass concentration inside the PBR that was integrated with different FO modules. During the operation with the side-stream osmotic system, the biomass concentration of *C. vulgaris* in the reactor was maintained at approximately 0.6 g/L. This indicates that the continuous supply of wastewater into the PBR allows the microalgae to grow at a rate to counterbalance the dilution effect caused by the influent wastewater. On the contrary, the biomass concentration kept increasing in the PBR integrated with the submerged FO module, reaching 0.965 g/L after 7 days of operation. The increment in biomass concentration is a direct result of the in-situ biomass dewatering with the submerged FO. If the experiment continues, a very concentrated microalgae biomass will be accumulated inside the PBR [36]. This would lead to a drop in microalgae growth due to the reduction in accessible light. This potential downside of the submerged FO-PBR can be controlled by increasing the influent flow rate, which, however, will reduce the hydraulic retention time and thus decrease the wastewater treatment efficiency in the PBR.

The FO module was applied to extract clean water out of the algae suspension and thus enhance the algae concentration (dewatering). Figure 4 presents the microalgae dewatering efficiency when *C. vulgaris* was harvested by side-stream and submerged FO modules. A lower values indicates more algal biomass lost during the FO process due to deposition onto the membrane surface [14]. It is clear that much less algal biomass loss was observed with the submerged FO compared with the side-stream configuration. Throughout the 7 days of operation, the algae dewatering efficiency was kept above 93.4%. The lower biofouling potential of microalgae in the submerged osmotic system may be due to the following reasons. First, the initial volume of algae suspension in the feed tank of the FO system was 1 L with side-stream configuration and 6 L with submerged configuration, respectively. At the end of each FO filtration, 750 mL of clean water was pulled through the membrane from algae suspension to the DS side. As a result, the biomass harvested with the submerged FO system is less concentrated in the feed tank. Second, the initial water flux with the submerged FO system is lower (more discussion is provided in Section 3.4). The less concentrated algae biomass and lower permeate drag force with the submerged FO system led to fewer bio-foulants being transported and then attached onto the membrane surface. With the side-stream configuration, algae dewatering efficiency was only 47% at the end of Day 1. The dewatering efficiency kept increasing and stabilized at ~70% on Day 7. This indicates that less algae deposition occurred onto the used membrane, which already had some algae deposited, compared to the fresh membrane. As the filtration progressed, the dewatering efficiency leveled off because the repulsive force between the suspended algae and the deposited algae on the membrane surface balanced the permeation drag force. The overall *C. vulgaris* dewatering efficiency by the side-stream FO module was obviously lower compared to our previously published study [15] where over 81% dewatering efficiency was achieved. However, it is difficult to provide a fair comparison of the FO dewatering efficiency between the two studies as (1) feed water chemistry (synthetic wastewater in the current study vs. BG-11 medium in the previous study [15]), (2) initial algae biomass concentration (0.6 g/L vs. 0.2 g/L) and (3) the FO membrane material (thin-film composite vs. cellulose triacetate) are different. A lower biomass concentration and cellulose triacetate membrane were found to have positive impacts on alleviating the biomass deposition and membrane fouling [29,37], and thus explain the higher *C. vulgaris* dewatering efficiency achieved with the side-stream FO module in the previous study [15].

### 3.3. Nutrients Removal in PBR Integrated with Different FO Configurations

Nitrogen and phosphorus play important roles in algal growth and metabolism [38]. As photoautotrophic microorganisms, microalgae utilize solar energy to assimilate nutrients from wastewater for cell synthesis and energy production within the algae cells [39]. The capacity of *C. vulgaris* to remove PO_4_-P, NH_3_-N and TN in the PBR integrated with different FO module configurations over 7 days of operation was examined, and the results were presented in Figure 5. In general, nutrients can be efficiently removed from the synthetic wastewater by *C. vulgaris*. The nutrients removal efficiency increased with time, indicating that (1) the microalgae became more and more adapted to the wastewater environment, and (2) the gradual salinity increase in the submerged FO-PBR did not show a negative impact on nutrients removal. After 7 days of the operation, 92.9%, 100% and 98.7% of PO_3_-P, NH_3_-N and TN were removed in the PBR integrated with the submerged FO module, and 82%, 96% and 94.8% of PO_4_-P, NH_3_-N and TN were removed in the PBR integrated with the side-stream FO module. The nutrients removal efficiency achieved in this study is consistent with those reported in the literature where high nitrogen removal (>91%) and phosphorus removal (>80%) by *C. vulgaris* were presented [40,41]. It is obvious that the removal of all nutrients (especially phosphate) was higher in the PBR integrated with the submerged FO, which could be attributed to the higher algae biomass concentration in the reactor. The uptake of nitrogen and phosphate by the microalgae cell involves an initial surface adsorption followed by internalization [42]. In the submerged FO-PBR, the more concentrated biomass provides more cell surface for nutrients adsorption to take place and thus achieves a higher nutrients removal rate. In addition to surface adsorption, the phosphate uptake rate also greatly depends on various stress factors, such as phosphorus deficiency [43]. In the PBR integrated with the submerged FO, phosphorous was relatively deficient due to the higher biomass in the reactor. This stressed condition would trigger a cascade of reactions to increase the saturated fatty acid synthesis and phosphorus consumption [43]. This may explain the larger difference in phosphate removal between the two configurations compared to those in other nutrients.

### 3.4. Water Flux Decline during Dewatering of C. vulgaris by FO Membrane

This section shows the water flux behavior during the FO filtration of the algae-treated wastewater. The FO water flux as a function of the permeate volume for both the side-stream and submerged configuration is summarized in Figure 6. As mentioned in the methodology, the baseline experiment was conducted before the algae suspension was filtered. The baseline flux is discussed first. Although the effective membrane area and solution compositions are the same with both FO configurations, the side-stream configuration had an obviously higher initial permeate flux (17.6 LMH) compared to the submerged configuration (12 LMH). A similar result was observed in a previous osmotic membrane bioreactor study [35] where the side-stream osmotic membrane bioreactor was reported to produce a much higher initial water flux compared to the submerged system. The different initial flux in the baseline experiments between the two configurations can be attributed to (1) the enhanced mass transfer and thus reduced external concentration polarization on the feed side of side-stream module due to crossflow [44], (2) the hydraulic pressure generated by the recirculation pump on the feed side of the side-stream module [30,35], and (3) air bubbles in the submerged module that may block the path for water to flow through. After 750 mL of clean water permeated through the membrane, little flux loss was observed (11.1% for the side-stream and 10% for the submerged configuration, respectively) in the baseline experiment. Thus, the severe flux decline during algae suspension filtration is attributed to membrane fouling.

For both configurations, the water flux showed a remarkable downward trend during the operation of each algae dewatering experiment. This was caused by the deposition of algae cells, cell fragments and their exudates onto the membrane surface thus forming a cake layer [14]. The thickness of the cake layer grew over time, leading to an augmented resistance for water to permeate [3]. In the side-stream configuration, a severe and rapid flux decline occurred. During the first 4 days, the flux decline rates almost remained stable and the values were relatively close, indicating a constant fouling onto the membrane surface throughout the experiment on each day. For example, the flux declined by 58.7% at the end of Day 1. Then the membrane was cleaned with deionized water flushing, which successfully swept away most of the cake layer from the membrane surface and recovered 90.3% of the water flux. On Day 2, the flux declined by 57.7% after filtration and the flux recovered to 89.5% after cleaning. On Day 3, the flux declined by 57.7% after filtration and the flux recovered to 91.2% after cleaning. On Days 5–7, the initial flux decline rate was comparable to the values on Days 1–4 but slowed down at the later filtration stage. Figure 7 summarizes the normalized flux loss at the start and end of the dewatering experiment on each day. The final flux losses were below 60% during the first 4 days but increased to 71.2% during the last 2 days. Although the side-stream configuration demonstrated a high fouling propensity, most of the flux decline was reversible by simple deionized water flushing without any chemical addition. After 6 days of operation, over 83% of the water flux could still be recovered. This indicates that the side-stream FO-PBR is suitable for long-term application.

In the submerged configuration, flux decline was moderate due to (1) relatively low permeate drag force and (2) slow increase of algae biomass in the PBR (feed tank of FO). The flux declined by 15.3% at the end of Day 1 and the final flux loss kept increasing, reaching 50% on Day 7 due to the solute and biomass build-up in the reactor. As no membrane cleaning was conducted between each filtration experiment, the flux loss at the start of each day followed a similar trend to the final flux loss (Figure 7).

Comparison between the two configurations showed that the final flux loss was more severe in the side-stream osmotic system because more foulants were brought to the membrane surface which is caused by (1) a higher permeate drag force and thus (2) greater biomass concentration over time. At the end of 7 days’ operation, the flux decline was 71.2% for the side-stream configuration and 50% for the submerged configuration, respectively. However, the initial flux loss was more noticeable in the submerged osmotic system especially after Day 2. Because no membrane cleaning was conducted in the submerged system, the algae biomass remained on the membrane surface, leading to flux loss accumulation.

## 4. Conclusions

This study investigated the role of FO module configurations (side-stream versus submerged) in the performance of osmotic PBR applied for nutrients removal and microalgae harvesting. The nutrients (nitrogen and phosphate) removal efficiency was higher in the PBR integrated with the submerged FO due to greater algae biomass in the reactor where in situ biomass dewatering was conducted. However, the continuous increase in algae biomass concentration inside the submerged FO-PBR will eventually lead to a decline in microalgae growth and thus nutrients removal when the biomass concentration is too high for the light to penetrate the medium and efficiently supply energy to the culture. Concerning the performance of the FO filtration, the side-stream configuration was demonstrated to be more efficient due to (1) higher initial permeate water flux and (2) easier fouling mitigation by simple hydraulic flushing. Although the higher initial permeate flux caused more algae biomass deposition onto the membrane surface and thus less dewatering efficiency, the deposited algae biomass can be easily scraped off the membrane [13], which will enhance the overall biomass recovery. The overall results suggest that a side-stream FO configuration allows a greater flexibility and is a more sustainable way for nutrients removal from wastewater and algae biomass harvesting.

## Figures and Tables

**Figure 1 membranes-12-00892-f001:**
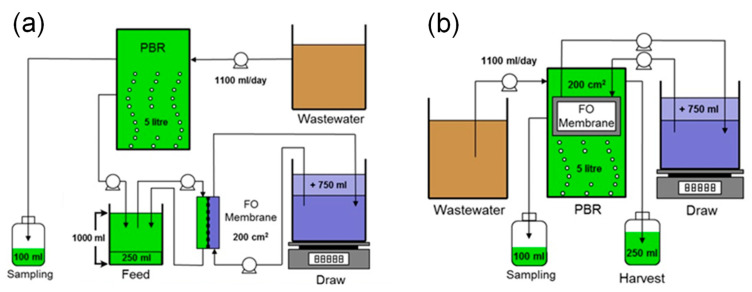
Schematic diagram of (**a**) side-stream and (**b**) submerged osmotic PBR.

**Figure 2 membranes-12-00892-f002:**
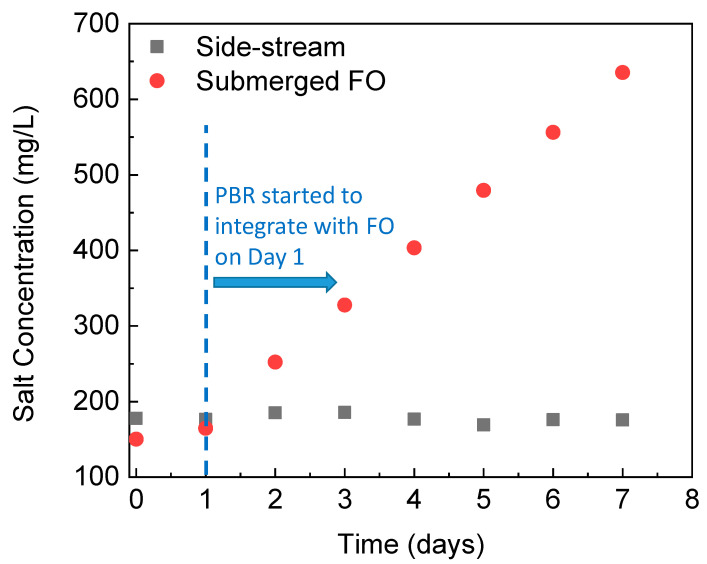
Changes in salinity in the PBR integrated with different FO configurations during a semi-continuous operation.

**Figure 3 membranes-12-00892-f003:**
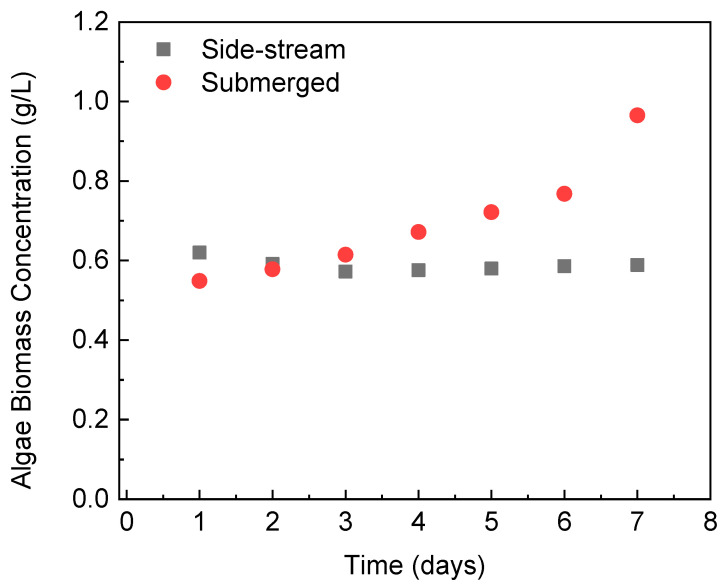
Algae biomass concentration in the PBR with different FO configurations during semi-continuous operation.

**Figure 4 membranes-12-00892-f004:**
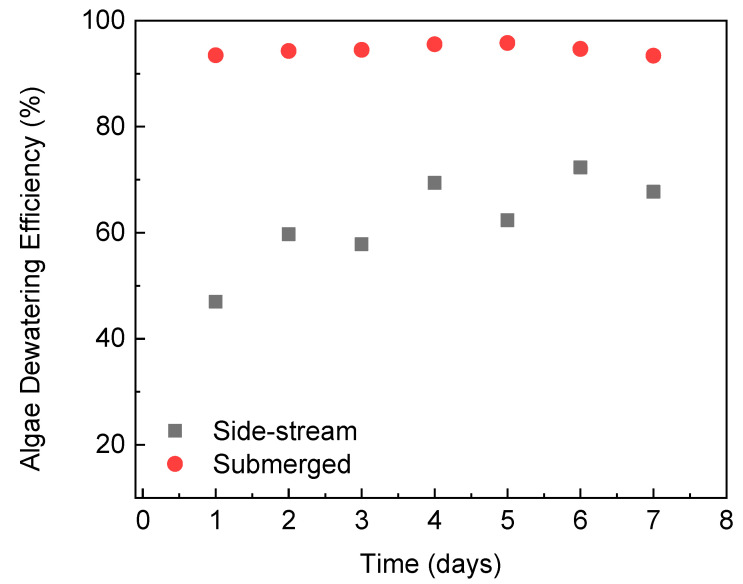
Algae dewatering efficiency at the end of each FO experiment when 750 mL of permeate was collected.

**Figure 5 membranes-12-00892-f005:**
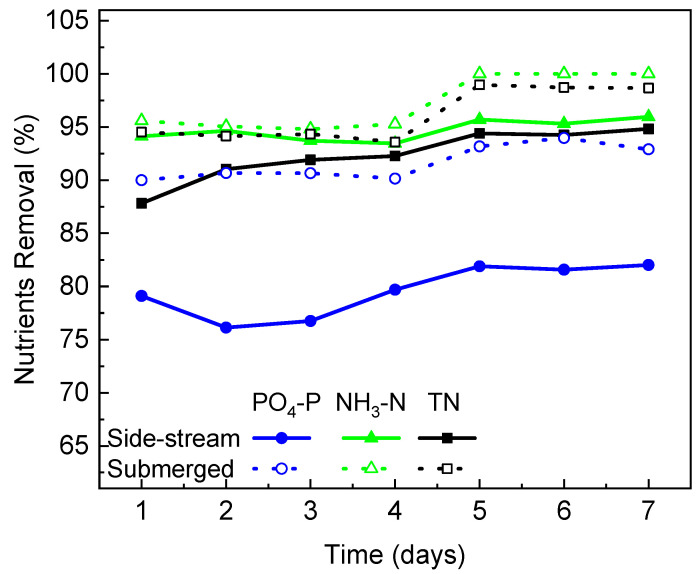
Comparison of nutrients removal efficiencies in the PBR with different FO configurations during semi-continuous operation.

**Figure 6 membranes-12-00892-f006:**
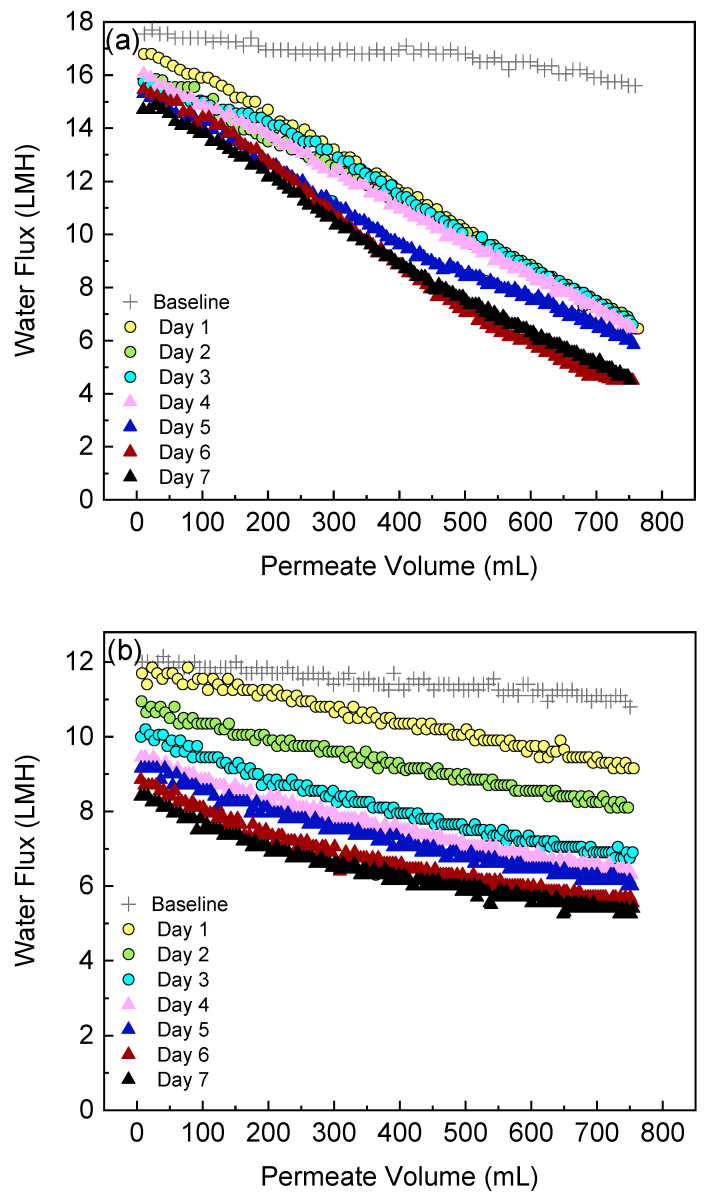
Change in permeate water flux during FO-PBR operation: (**a**) side-stream and (**b**) submerged FO configurations.

**Figure 7 membranes-12-00892-f007:**
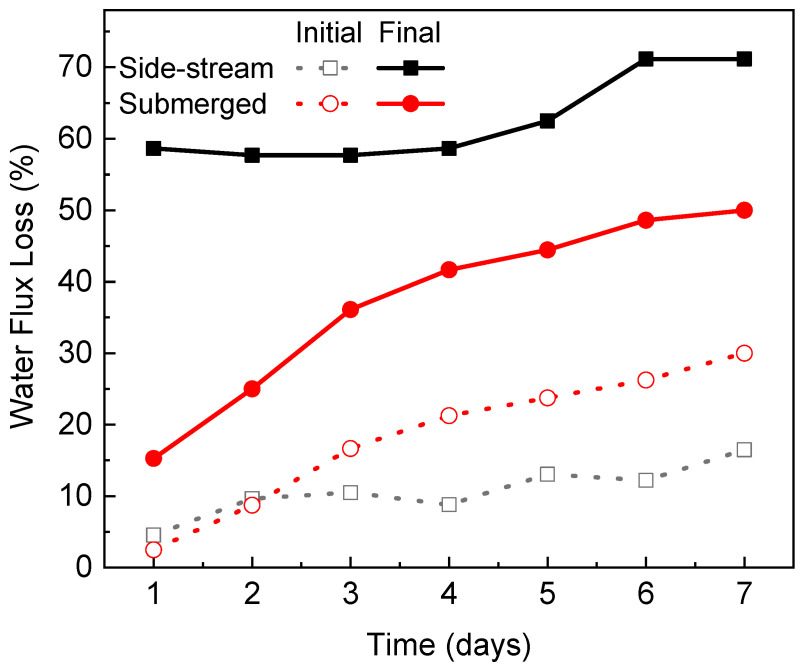
Normalized flux loss at the start and end of each algae dewatering experiment.

**Table 1 membranes-12-00892-t001:** Recipe of synthetic wastewater.

Component	Concentration (mg/L)
K_2_HPO_4_∙3H_2_O	30
CaCl_2_∙2H_2_O	7.5
Cr(NO_3_)_3_∙9H_2_O	1.125
CuCl_2_∙2H_2_O	0.75
MnSO_4_∙H_2_O	0.15
NiSO_4_∙6H_2_O	0.375
PbCl_2_	0.15
ZnCl_2_	0.375
Urea	120
NH_4_Cl	15
CH_3_COONa∙3H_2_O	168.75
Peptone	22.5
MgHPO_4_∙3H_2_O	37.5
FeSO_4_∙3H_2_O	7.5
Starch	157.5
Milk powder	150
Dried yeast	67.5
Soy oil	37.5

## Data Availability

The data presented in this study are available on request from the corresponding author.

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
