# Peer review of "The Influence of Forward Osmosis Module Configuration on Nutrients Removal and Microalgae Harvesting in Osmotic Photobioreactor"

_membranes, 2022, doi:10.3390/membranes12090892_

Round 1
Reviewer 1 Report
In this manuscript, two FO module configurations were integrated with photobioreactor in order to evaluate microalgal nutrient removal, algae harvesting efficiency and membrane fouling. However, the whole experimental period is short and it is difficult to determine the type of membrane contamination. It is suggested that the authors further study this problem in future work and propose an effective continuous operation method of the process. Overall, it is really an interesting topic.
Here, I suggest a major revision.
Title
The title is not so representative of this work. In fact, the effect of FO-PBR configuration on the system performance was abstained.
Keywords
Some keywords appear in the title, I suggest author replace them.
2. Materials and Methods
Line 108, the most suitable species?
Line 114 why is it cited here? Instrument and its information should be mentioned.
Line 119 why cite here? Chemicals’ manufacturer can be mentioned.
Table 1, why the synthetic wastewater was prepared as the described condition? Is there any reference, especially the acetate, Urea, ammonium-N and P concentration? Why was acetate used as carbon source? Any reference?
If there was no such wastewater, authors would not consider the effect of algae treatment on wastewater, but to analyse the nutrient utilization efficiency.
Lines 129-133 why cite here?
3. Results and Discussion
It seems like a research report. Here, a lot of data were provided without an effective discussion. Especially, most of the results were not compare with that of the public articles, lack of convincing.
Reference:
The references cited should reflect the new achievements and latest progress in this field, and it is suggested to cite more literatures in the last five years.
Reviewer 2 Report
The author reported a study to evaluate the sewage concentration performance of FO membranes in different modes. The study is designed very well, and the key findings are well supported by the results. I recommended this manuscript be published in Membranes after solve the minor issues:
1. Figure 6. Move the label (a) and (b) to the top left corner of each figure.
2. Figure 5. The difference between PO4-P removal rate in side-stream and submerged modes is larger than those in other nutrients. Any specific reason for this result?
Round 2
Reviewer 1 Report
The manuscript has been well revised.